# The prevalence of psychiatric symptoms before the diagnosis of Parkinson's disease in a nationwide cohort: A comparison to patients with cerebral infarction

Szabolcs Szatmári, Jr[1,2,3], András Ajtay[2,3], Ferenc Oberfrank[4], Balázs Dobi[3,5], Dániel Bereczki[2,3] *

1 János Szentágothai Doctoral School of Neurosciences, Semmelweis University, Budapest, Hungary, 2 Department of Neurology, Semmelweis University, Budapest, Hungary, 3 MTA-SE Neuroepidemiological Research Group, Budapest, Hungary, 4 Institute of Experimental Medicine, Budapest, Hungary, 5 Department of Probability Theory and Statistics, Eötvös Loránd University, Budapest, Hungary

* bereczki@neur.sote.hu

## Abstract

### Objectives

Psychiatric symptoms (PS) can be non-motor features in Parkinson's disease (PD) which are common even in the prodromal, untreated phase of the disease. Some PS, especially depression and anxiety recently became known predictive markers for PD. Our objective was to explore retrospectively the prevalence of PS before the diagnosis of PD.

### Methods

In the framework of the Hungarian Brain Research Program we created a database from medical and medication reports submitted for reimbursement purposes to the National Health Insurance Fund in Hungary, a country with 10 million inhabitants and a single payer health insurance system. We used record linkage to evaluate the prevalence of PS before the diagnosis of PD and compared that with patients with ischemic cerebrovascular lesion (ICL) in the period between 2004–2016 using ICD-10 codes of G20 for PD, I63-64 for ICL and F00-F99 for PS. We included only those patients who got their PD, ICL and psychiatric diagnosis at least twice.

### Results

There were 79 795 patients with PD and 676 874 patients with ICL. Of the PD patients 16% whereas of those with ischemic cerebrovascular lesion 9.7% had a psychiatric diagnosis before the first appearance of PD or ICL (p<0.001) established in psychiatric care at least twice. The higher rate of PS in PD compared to ICL remained significant after controlling for age and gender in logistic regression analysis. The difference between PD and ICL was significant for Mood disorders (F30-F39), Organic, including symptomatic, mental disorders

**Funding:** -DB was supported by grants from the National Brain Research Program, nr. 2017-2-1 NKP-2017-00002 (https://nkfih.gov.hu/palyazoknak/nkfi-alap/nemzeti-kivalosagi) -SS has recieved grants from: the Higher Education Institutional Excellence Program and the New National Excellence Program (UNKP-17-3) of the Ministry of Human Resources of the Government of Hungary (http://semmelweis.hu/innovacio/2017/05/08/doktori-hallgato-doktorjelolt-unkp-17-3/) and the EFOP-3.6.3-VEKOP-16-2017-00009 project for development of scientific workshops for medical, health sciences and pharmaceutical training (http://semmelweis.hu/innovacio/palyazat/tamogatott-projektek/hazai-tamogatott-projektek/efop-3-6-3-vekop-16-2017-00009/) The funders had no role in study design, data collection and analysis, decision to publish, or preparation of the manuscript.

**Competing interests:** The authors have declared that no competing interests exist.

(F00-F09), Neurotic, stress-related and somatoform disorders (F40-F48) and Schizophrenia, schizotypal and delusional disorders (F20-F29) diagnosis categories (p<0.001, for all).

## Discussion

The higher rate of psychiatric morbidity in the premotor phase of PD may reflect neurotransmitter changes in the early phase of PD.

## Introduction

Parkinson's disease (PD) is a gradual-onset multisystem neurodegenerative disorder, currently clinically defined by a set of classical motor signs. However, converging evidence from neuropathological, clinical, and imaging research suggests a definable prodromal phase where detectable non-motor features are major components [1–3]. Psychiatric symptoms (PS) are some of the non-motor features which are common even in the prodromal, untreated phase of PD, may precede motor symptoms and have a major impact on the course of the disease, quality of life and caregivers [4, 5]. Some PS, especially depression and anxiety recently became known predictive markers for PD with several prospective studies reporting diagnostic value [3]. To explore PS that are related to PD itself it is necessary to study the prodromal phase and untreated patients as antiparkinsonian drugs have the potential to influence cognition, affective, psychotic and impulsivity symptoms. We recently reviewed the literature on PS addressing the early, untreated stages of PD, with a focus on the premotor and early motor phases separately and found that the prevalence of PS is high even in the premotor/prodromal phase [6]. The onset and characteristics of PS in drug naïve patients before the diagnosis of PD are still not well studied, further examination of patients in the premotor, untreated phase is needed to better understand PS which are linked to the disease pathology itself. Recently, we demonstrated that incidence and prevalence rates of PD are considerably higher than previous reports in Hungary and the healthcare administrative database of the National Health Insurance Fund (NHIF), using the International Classification of Diseases 10th revision (ICD-10), with proper case identification and certification methodology is appropriate to evaluate epidemiological features of PD in Hungary [7]. Therefore, using this database with a massive sample size our objective was to explore retrospectively the prevalence of PS before the diagnosis of PD in the Hungarian population.

## Materials and methods

We identified PD patients with PS using the database of the NHIF which includes data from specialist outpatient and specialist inpatient services of all hospitals covering the whole population of Hungary. The NHIF does not include data from general practitioners. Encrypted identifiers were used as the original patient identifier codes were anonymized. The database contains information of PD patients covering a 13-year period between 2004 and 2016. Case ascertainment and patient selection are described in detail in a previous paper [7]. First, patients who used either the inpatient or the outpatient specialist health care service at least once during this period and had a primary or secondary diagnosis of PD (ICD-10, G20) were identified. We considered those patients to have PD who appeared at least in two calendar years with G20 code in the database during the 13-year period regardless of the diagnosis type (i.e., primary/admitting diagnosis or secondary diagnosis). This is the PD patient group that was further analyzed. Third, the range and prevalence of all mental and behavioral disorders

(ICD-10, codes F00-F99) were analyzed in the period before the first appearance of PD in the database. For more conservative case ascertainment the F00-F99 diagnoses had to be established at least twice in an outpatient or inpatient psychiatric care service. For the control group we chose all patients with ischemic cerebrovascular lesion (ICL) diagnoses (ICD-10, codes I63 or I64) assigned at least twice between 2004 and 2016. In this group, patients with PD or parkinsonism (ICD-10, codes G20-26) with any diagnosis type (primary/secondary) were not included. In both groups (PD and ICL) patients below 40 and above 100 years of age (age in the year of the first PD or ICL diagnosis) were excluded as well as patients who refilled any antiparkinsonian drug (APD) before their first PD or ICL diagnosis. Pharmacological data is available only from 2010, therefore our analysis could evaluate prescriptions of APDs refilled at pharmacies only from 2010 onwards and had no access to data on inpatient medication use in hospitals.

Personal data protection regulations were followed, the Ethics Committee of Semmelweis University, Budapest, Hungary approved the study (Approval No: SE TUKEB 88/2015).

Data extraction and results of individual searches in the database were exported to excel files. This was performed by a research assistant IT specialist, with several years of experience in reviewing medical records of patients with neurological conditions. The data from excel files were first analyzed by basic descriptive statistical methods. For associations between the PD and ICL group we used chi-squared test, logistic regression and mixed effects logistic regression analysis. The goodness of fit testing of the traditional logistic regression models was considered using the Hosmer-Lemeshow test but was eventually not used due to the large sample size, which resulted in significant test outcomes even when there was only a minor deviation. R version 3.6.2 was used for data analysis with packages lme4 and Resource selection.

## Results

The number of patients between 2004–2016 with at least one PD diagnosis (G20) in the specialist healthcare service (outpatient and inpatient) was 130 773. Fulfillment of the minimum 2-year PD diagnosis criteria was present in 79 795 patients, this was the cohort for further analysis. The mean ± SD age of this group was 73 ± 9 years, 48% were male. Of the 79 795, 59 403 patients had at least one mental and behavioral disorder (F00-F99) anytime during the examined 13 year period. The number of patients who had their F00-F99 diagnosis before the first PD diagnosis was 26 645. Out of the 26 645 patients 13 092 had their F00-F99 diagnosis in specialized psychiatric care (outpatient/inpatient) established at least twice.

The control group between 2004 and 2016 with at least one ICL diagnosis (I63 or I64) and without PD diagnosis consisted of 783 843 patients. Of those 676 874 got their ICL diagnosis at least twice. The mean ± SD age of this group was 69 ± 12 years, 46% were male. 330 170 patients with ICL had at least one F00-F99 diagnosis during the studied 13 years. Of those, 160 501 had their psychiatric diagnosis before the diagnosis of ICL, and out of the 160 501 patients 66 164 had their F00-F99 diagnosis established in psychiatric care at least twice. Fig 1 shows the algorithm used for cohort definition.

To evaluate the overall presence of psychiatric morbidity, in addition to the number of patients we also analyzed the total number of reported F00-F99 diagnoses, noting, that an individual patient may have several diagnoses in the F00-F99 categories. The numbers of all F00-99 diagnoses before the initial PD and ICL diagnosis were 63 360 vs. 340 280, and of those established in psychiatric care were 25 674 vs. 123 746 for PD and ICL, respectively. Table 1 shows the number of patients and their mean age at the first diagnosis. Table 2 presents the number of diagnoses that were established in psychiatric care at least twice, their proportion in

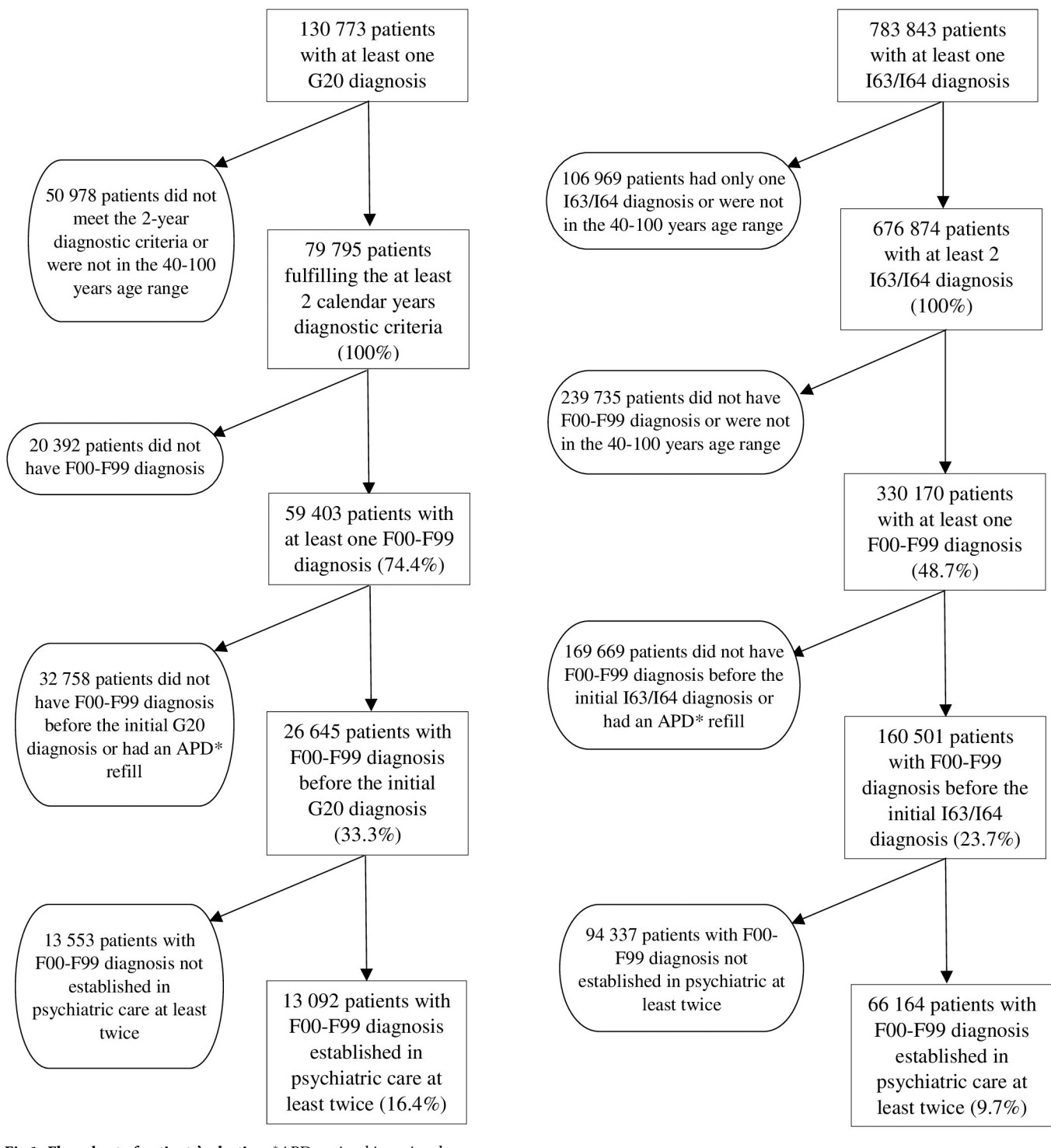

**Fig 1. Flow chart of patients' selection.** *APD-antiparkinsonian drug.

all 79 795 PD and all 676 874 ICL patients and the effect of age and gender. Table 3 presents the age of patient subgroups by F-categories.

Of the PD patients 16% whereas of those with ICL 9.7% had a psychiatric diagnosis before the first appearance of PD or ICL (p<0.001). The higher rate of PS in PD compared to ICL

**Table 1. Age of the patient subgroups.**

| Patient subgroup | PD (n) | Mean age (±SD) at initial appearance of G20 | ICL (n) | Mean age (±SD) at initial appearance of I63-I64 |
|---|---|---|---|---|
| All patients | 79 795 | 73.2 ± 9.0 | 676 874 | 69 ± 12 |
| Patients with F00-99 dg. | 59 403 | 72.9 ± 9.2 | 330 170 | 67.7 ± 12.1 |
| Patients with F00-99 dg. before PD/ICL dg. | 26 645 | 72.1 ± 9.5 | 160 501 | 66.4 ± 12.2 |
| Patients with F00-99 dg. established in psychiatric care before PD/ICL dg. | 13 092 | 71.3 ± 10 | 66 164 | 65.1 ± 12.3 |

remained significant after controlling for age and gender in logistic regression analysis (p<0.01 for age, gender and diagnosis type). We found that females had greater odds compared to males, and older patients had lowers odds of preceding psychiatric diagnoses. In the PD group where psychiatric diagnosis appeared before PD diagnosis, in descending order Mood disorders (F30-F39: 9%), Organic, including symptomatic, mental disorders (F00-F09: 9%), Neurotic, stress-related and somatoform disorders (F40-F48: 7%) and Schizophrenia, schizotypal and delusional disorders (F20-F29: 4%) were the most common diagnosis categories, compared with only 6% (p<0.001), 4% (p<0.001), 5% (p<0.001) and 1% (p<0.001) in the ICL group in the same categories.

The number of psychiatric diagnoses in total and for the sub-diagnoses were also analyzed using traditional logistic and mixed effects logistic regression analysis. The traditional models were used simply on the psychiatric diagnoses as an outcome variable while the mixed effects model also took into account that several diagnoses may belong to the same patient. Analyses were conducted with and without controlling for the effect of age and gender.

Without controlling and with considering all types of psychiatric diagnoses together we found that the odds of receiving a psychiatric diagnosis before a PD/ICL diagnosis is significantly greater for the PD group (p<0.001). This was true in both types of logistic regression models. The effect of PD diagnosis in the different psychiatric diagnosis subgroups was varying.

When we adjusted for age and gender the results were much more consistent: the odds of preceding any (OR: 1.429 [95% C.I: 1.395–1.464]), or F00-F09, F20-F29, F30-F39, F40-F48, F50-F59, F60-F69 or F70-F79 diagnoses were significantly greater for the PD group of patients in both types of logistic regression models (p<0.001 for all comparisons). The only exception was F10-F19 (mental and behavioral disorders due to psychoactive substance use), where the ICL group had greater odds of preceding psychiatric diagnosis. Logistic regression could not be conducted for the sub-diagnoses of F80-89 and F99 due to lack of relevant diagnoses. The adjusted model for the diagnoses F90-98 also had a relatively low amount of diagnoses to work with, and while the regressions could be conducted, they did not produce any significant effect for the PD/ICL groups. The multiple testing might warrant for correction, however the tests were not directly comparable (as they used different sub-diagnoses i.e. different outcome variables) and the p-values were so low that the effects would remain significant even after correction, thus our overall conclusions would not change. Summing up the effect of gender and age, generally we found that females had greater odds compared to males, and older patients had lowers odds of preceding psychiatric diagnoses. The results of the mixed effects multiple logistic regression models are shown in Table 2. We also conducted a sensitivity analysis where we only kept the data of such psychiatric diagnoses where medication information was also available. The results of the controlled mixed effects logistic regression models were very close to

**Table 2. Number of F00-F99 diagnoses which were established in psychiatric care before PD and ICL diagnoses between 2004–2016 in the ICD-10 subgroups and the effect of the presence of PD vs. ICL, age and gender on F00-99 diagnoses.**

| ICD code | F diagnostic subgroup | Number of F diagnoses before G20, established in psychiatric care (number of patients) | Diagnoses established in psychiatric care relative to all PD patients (n = 79 795) | Number of F diagnoses before I63/64 established in psychiatric care (number of patients) | Diagnoses established in psychiatric care relative to all ICL patients (n = 676 874) | PD relative to ICL | | Age | | Gender (female) | |
|---|---|---|---|---|---|---|---|---|---|---|---|
| | | | | | | Odds ratio (95% C.I.) | p-value | Odds ratio (95% C.I.) | p-value | Odds ratio (95% C.I.) | p-value |
| F00-F09 | Organic, including symptomatic, mental disorders | 7361 (5594) | 9% | 24 656 (19 961) | 3% | 1.469 (1.425–1.514) | <0.001 | 1.033 (1.032–1.034) | <0.001 | 1.202 (1.172–1.233) | <0.001 |
| F10-F19 | Mental and behavioral disorders due to psychoactive substance use | 584 (569) | 1% | 6768 (6620) | 1% | 0.665 (0.610–0.724) | <0.001 | 0.950 (0.948–0.952) | <0.001 | 0.275 (0.261–0.290) | <0.001 |
| F20-F29 | Schizophrenia, schizotypal and delusional disorders | 2914 (2387) | 4% | 6981 (5744) | 1% | 2.390 (2.273–2.514) | <0.001 | 0.986 (0.985–0.988) | <0.001 | 1.563 (1.492–1.638) | <0.001 |
| F30-F39 | Mood (affective) disorders | 7574 (5982) | 9% | 40 684 (32 167) | 5% | 1.281 (1.247–1.317) | <0.001 | 0.967 (0.966–0.968) | <0.001 | 2.121 (2.076–2.168) | <0.001 |
| F40-F48 | Neurotic, stress-related and somatoform disorders | 5897 (5170) | 7% | 36 511 (31 405) | 5% | 1.127 (1.095–1.161) | <0.001 | 0.964 (0.963–0.964) | <0.001 | 2.091 (2.045–2.137) | <0.001 |
| F50-F59 | Behavioral syndromes associated with physiological disturbances and physical factors | 461 (460) | 1% | 2227 (2220) | 0% | 1.381 (1.247–1.530) | <0.001 | 0.969 (0.966–0.972) | <0.001 | 2.125 (1.950–2.314) | <0.001 |
| F60-F69 | Disorders of adult personality and behavior | 557 (536) | 1% | 4262 (4104) | 1% | 1.188 (1.085–1.301) | <0.001 | 0.927 (0.924–0.929) | <0.001 | 1.409 (1.329–1.495) | <0.001 |
| F70-F79 | Mental retardation | 268 (229) | 0% | 1379 (1250) | 0% | 1.684 (1.456–1.947) | <0.001 | 0.924 (0.920–0.929) | <0.001 | 1.004 (0.906–1.113) | 0.935 |
| F80-F89 | Disorders of psychological development | 0 (0) | 0% | 6 (6) | 0% | NA | NA | NA | NA | NA | NA |
| F90-F98 | Behavioral and emotional disorders with onset usually occurring in childhood and adolescence | 58 (58) | 0% | 269 (268) | 0% | 1.203 (0.903–1.602) | 0.207 | 1.002 (0.993–1.011) | 0.643 | 2.221 (1.723–2.862) | <0.001 |
| F99 | Unspecified mental disorder | 0 (0) | 0% | 3 (3) | 0% | NA | NA | NA | NA | NA | NA |
| | All F-ICD type in all PD (79 795) and ICL (676 874) patients | 25 674 (13 092) | 32% | 123 746 (66 164) | 18% | 1.358 (1.338–1.378) | <0.001 | 0.976 (0.976–0.977) | <0.001 | 1.736 (1.706–1.767) | <0.001 |

*: An individual patient may appear repeatedly in an F subgroup and also in several F subgroups if that patient had been assigned multiple F codes.

the original ones. However, due to the severely decreased number of patients in the sensitivity analysis the effect of PD/ICL diagnosis lost its significance on the preceding F10-19, F50-59, F60-69, F70-79 sub-diagnoses.

Fig 2A presents the distribution of F00-F99 diagnoses in proportion of all PD (79 795) and ICL (676 874) patients, and Fig 2B shows the distribution of diagnoses which were established in specialized psychiatric care. In both figures an individual patient may have appeared repeatedly in an F subgroup and also in several F subgroups if that patient had been assigned multiple F codes. The average time when psychiatric diagnoses appeared before PD diagnosis was 3.1 years in the PD group and 4 years in ICL.

**Table 3. Age of patients for F00-99 diagnosis categories between 2004–2016 for those where F diagnoses were assigned at least twice by psychiatric care facilities.**

| ICD code | Title | Mean age at F00-99 dg. in PD (±SD) | N* | Mean age (±SD) at initial appearance of G20 | Mean age at F00-99 dg. in ICL (±SD) | N* | Mean age (±SD) at initial appearance of I63-I64 |
|---|---|---|---|---|---|---|---|
| F00-09 | Organic, including symptomatic, mental disorders | 72.7 ± 9 | 7096 | 74.8 ± 8.7 | 69.9 ± 12.5 | 24 656 | 72.3 ± 12.1 |
| F10-19 | Mental and behavioral disorders due to psychoactive substance use | 62 ± 9.7 | 584 | 65.6 ± 9.2 | 55.1 ± 9.7 | 6768 | 59.5 ± 9.3 |
| F20-29 | Schizophrenia, schizotypal and delusional disorders | 65.4 ± 11.3 | 2914 | 68.2 ± 10.8 | 62.1 ± 13.2 | 6981 | 66 ± 12.5 |
| F30-39 | Mood (affective) disorders | 66.1 ± 10.4 | 7574 | 69.5 ± 10 | 58.5 ± 11.4 | 40 684 | 62.7 ± 11.3 |
| F40-48 | Neurotic, stress-related and somatoform disorders | 65.9 ± 10.6 | 5897 | 69.3 ± 10.2 | 58 ± 11.7 | 36 511 | 62.2 ± 11.5 |
| F50-59 | Behavioral syndromes associated with physiological disturbances and physical factors | 66.4 ± 10.3 | 461 | 69.1 ± 10 | 59.6 ± 11.4 | 2227 | 63 ± 11.3 |
| F60-69 | Disorders of adult personality and behavior | 60.6 ± 10.2 | 557 | 64.5 ± 9.9 | 52.4 ± 9.1 | 4262 | 56.7 ± 9.2 |
| F70-79 | Mental retardation | 54.1 ± 10.4 | 268 | 58 ± 10.2 | 52 ± 9.8 | 1379 | 56 ± 9.4 |
| F80-89 | Disorders of psychological development | no data | 0 | no data | 49 ± 4.9 | 6 | 51.8 ± 4.3 |
| F90-98 | Behavioral and emotional disorders with onset usually occurring in childhood and adolescence | 70.8 ± 11 | 58 | 73.3 ± 10.4 | 64.7 ± 14.1 | 269 | 68 ± 13.2 |
| F99 | Unspecified mental disorder | no data | 0 | no data | 49.6 ± 5.5 | 3 | 56 ± 2 |

*: An individual patient may appear repeatedly in an F subgroup and also in several F subgroups if that patient had been assigned multiple F codes.

## Discussion

By analyzing the large healthcare administrative database of the NHIF our aim was to estimate the prevalence of mental and behavioral disorders before the diagnosis of PD. Our primary findings were that several psychiatric diseases (especially mood disorders, cognitive impairment, anxiety disorders, schizophrenia) were more common in PD patients before PD diagnosis compared to our control group, patients with ICL. If we consider only those diagnoses that were established by psychiatric care, the numbers are still significantly higher in PD than in the control group. Our results are in line with previous findings and support the notion that a range of PS are already present at the diagnosis of PD, and these symptoms are more common in PD patients than in people without PD [4, 8]. Henceforth, we will discuss below only the most common PS that we have found.

Our data indicate that from the Organic, including symptomatic, mental disorders group (F00-09) category the diagnosis rate in PD patients was 9%, most of which were different types of dementias, showing evidence of cognitive dysfunction. In comparison, only 4% of those with ICL had diagnoses from the F00-09 group. It was previously demonstrated that the prevalence of cognitive impairment is high even in newly diagnosed PD patients, it is associated with extensive atrophy and greater percentage of cortical thinning, older age at disease onset is an important determinant of cognitive dysfunction [9–11] and up to 80% of all PD patients will eventually develop dementia [12]. Our results support these findings because in our PD sample the average age in the year of the appearance of the first PD diagnosis in our database was relatively high, 73 years. Although we do not have information on detailed

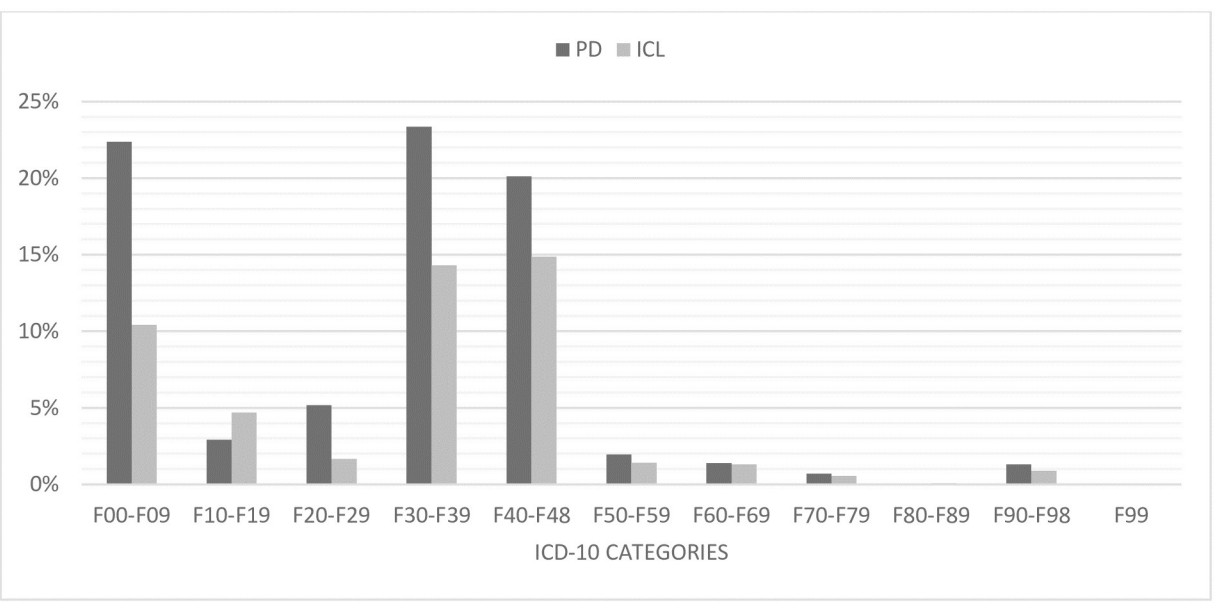

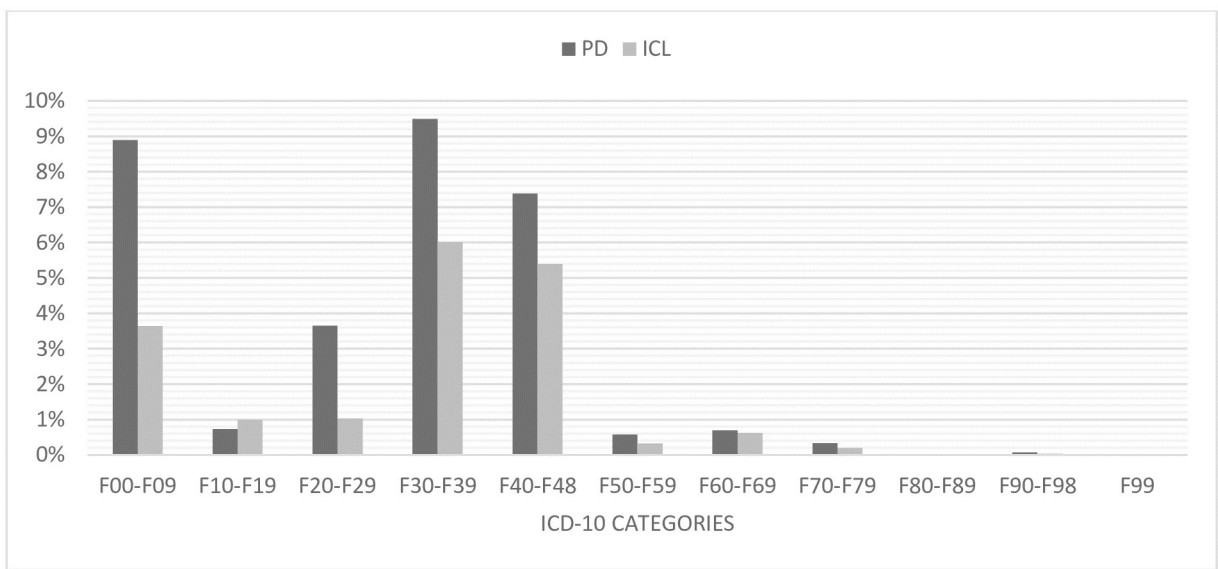

**Fig 2.** (A) Distribution of F00-F99 diagnoses in proportion of all patients with PD and ICL. (B) Distribution of F00-F99 diagnoses established in psychiatric care at least twice, in proportion of all patients with PD and ICL. *: an individual patient may appear repeatedly in an F subgroup and also in several F subgroups if that patient had been assigned multiple F codes.

neuropsychological results, we assume that the presence of the large number of dementia diagnoses may be due to the relatively older age of the patients. PD typically occurs in people over the age of 60 [13]. It has to be acknowledged that the mean age at the first appearance of G20 diagnosis in our dataset was at the higher end of those in the literature. However, in a French study which assessed PD with a similar methodology, using the national healthcare insurance database, incidence rates were the highest starting from the 74 age group [14]. Additionally, a study conducted in Germany, which examined prevalence and incidence based also on health claims data, the mean age of PD onset was 77 years [15]. A systematic review of incidence

studies of PD found that PD incidence rises steadily with age to a peak occurring between the ages of 70 to 79 [16]. The differences in large health claim data analyzes can be explained by different validation strategies and in our case by the fact that we only considered those with first appearance of G20 from a defined time point, i.e. starting from 2004. Therefore it is obvious that there are many cases that had been diagnosed with PD earlier than the database was initiated, and the first appearance of these patients in the database appears long after their first clinical diagnosis of PD.

We found significant difference also in the Schizophrenia, schizotypal and delusional disorders (F20–F29) category, where schizophrenia diagnosis (F20) was the most common. In the PD group there was a more than twofold higher prevalence (4%) compared to the control group (1%). A study based on similar methods as ours, analyzing ICD codes retrospectively in a large health insurance database investigated the effects of psychiatric diseases on subsequent PD diagnosis in an Asian population and found that having a specific psychiatric disease was independently associated with a nearly 2.3-fold increased risk of receiving a PD diagnosis within the 6-year follow-up period, patients with schizophrenia exhibiting the highest risk of a later PD diagnosis [17]. Our results support this result, however the exact association between schizophrenia and PD remains to be elucidated, because there are still just a few studies with inconclusive data which examined the connection between the two diseases. The role of dopamine and dopaminergic pathways that play crucial role in both PD and schizophrenia is well-known [18]. Current data assume that negative and cognitive symptoms in schizophrenia anticipate the onset of positive symptoms, decreased dopamine may be responsible for these, suggesting that schizophrenia may essentially represent a hypodopaminergic disorder after all [19, 20]. On the other hand a few studies have proposed the idea that parkinsonism in schizophrenia may not be just the consequence of neuroleptic exposure but it could be also related to the clinical spectrum and might represent the motor side of schizophrenia manifesting in the late phase of disease course [21–23].

In the Mood (affective) disorders (F30–F39) category the diagnosis rate was higher in the PD sample (9%) and it was significantly larger than in the ICL group (6%). One of the most frequent disorder before PD diagnosis was, not surprisingly the depressive episode in both PD and ICL groups. Depressive symptoms predating PD can have diverse etiologies, although the exact pathomechanism is still unknown. Neurobiological factors, changes in brain structure and neuronal systems (dopaminergic, noradrenergic, and serotonergic) that are associated with the underlying neurodegenerative processes in PD play an important role in the early stages, later on mood reaction to the progressive disability and PD itself, psychosocial factors or pain could also have roles in the appearance of depressive symptoms [24, 25]. Several other studies came to the conclusion that depression may also be an independent risk factor for PD [26, 27]. Our prevalence values correlate favorably with previous findings [6] and further support the idea of depression being a non-motor symptom already present before the diagnosis of PD.

Another group which showed significant difference was the Neurotic, stress-related and somatoform disorders (F40–F48) category, before PD diagnosis being present in 7%, while before ICL only in 5%, anxiety disorders being the most common in this category. The etiology of anxiety in PD is also multifactorial, a dysfunction in the dopaminergic system might be implicated from the earliest stages of disease [28] likewise decreased metabolism and tissue reduction in different brain areas may also contribute to the development of anxiety [29, 30]. Moreover, the risk of developing PD was found to be higher in a population with anxiety, also the psychological reaction to the development of motor disability and other common anxiety disorders described in PD like panic attacks and social phobias have an important role [31,

32]. Our study provides further evidence on anxiety disorders having a high prevalence before PD diagnosis.

The average number of years when psychiatric diagnoses appeared before PD diagnosis, which was 3.1 years in our case, slightly differ from other study findings about prodromal symptoms such as depression and anxiety that emerged 4–6 years before PD motor symptoms appeared [33]. This lag time may correspond to the proposed beginning of the disease process that anticipates the onset of motor symptoms by several years or even decades [34] and the difference can be explained by the relatively shorter examination period in our study.

The strength of our study lies in the large-scale epidemiologic examination and sufficient number of cases for valid statistical comparison. The database analysis of a several years' time period with full coverage of a country makes this study large about PD and psychiatric comorbidity. The use of all categories of discharge diagnoses types (admitting diagnosis, principal and secondary diagnoses) according to the ICD-10-CM Official Guidelines for Coding and Reporting increases the efficacy of case identification both in PD and psychiatric illnesses.

We are aware that our research may have several limitations, therefore our results should be interpreted within the context of these. First, diagnostic accuracy may be limited in each patient group (PD, psychiatry, ICL) due to the lack of a direct individual clinical case certification by a specialist in the field. Second, using health administrative databases, the possibility of misclassification error should be considered as an important source of research bias which threatens the validity of study results. However it was demonstrated that databases created from hospital reports submitted for reimbursement purposes can be used reliably in Hungary for ICL (I63-I64) epidemiological studies [35] and as previously mentioned, with appropriate case identification methodology this is true for PD as well [7]. Additionally, patients not presenting in specialized outpatient or inpatient care but only at general practitioners, patients who have not yet been commenced on APD, undiagnosed patients, result in underestimation of the number of patients with PD. At the same time, all this highlights the importance of treating PS from the onset of PD, taking advantage of the benefits of certain APDs, but also being cautious of potential side effects (e.g. psychosis). That is why it is important that the diagnosis and treatment of PD to be managed from the beginning by a neurologist specialist, as it is the case in Central and Eastern Europe [36, 37]. Third, information on APD use is incomplete. This could have also influenced the results nonetheless a paper reports that the majority of PS do not worsen substantially in the first years of PD regardless of treatment, and the initiation of antiparkinsonian therapy does not significantly improve most of these symptoms on average [8]. Despite these drawbacks, our study provides up till now not available data regarding the current epidemiology of the psychiatric burden experienced in this large group of PD patients.

In conclusion, we determined that a range of psychiatric illnesses have elevated prevalence before the first diagnosis of PD compared to ICL. This study may reflect neurotransmitter changes in the early phase of PD and highlights the importance of early, routine screening for highly prevalent, often under-diagnosed and under-treated PS in PD patients to initiate optimal treatment.

## Supporting information

**S1 File.**
(XLSX)

**S2 File.**
(XLSX)

**S3 File.**
(XLSX)

## Author Contributions

**Conceptualization:** Szabolcs Szatmári, Jr, Dániel Bereczki.

**Data curation:** Szabolcs Szatmári, Jr, András Ajtay, Ferenc Oberfrank, Dániel Bereczki.

**Formal analysis:** Szabolcs Szatmári, Jr, Balázs Dobi, Dániel Bereczki.

**Investigation:** Szabolcs Szatmári, Jr, Balázs Dobi.

**Methodology:** Szabolcs Szatmári, Jr, András Ajtay, Balázs Dobi, Dániel Bereczki.

**Resources:** András Ajtay, Ferenc Oberfrank, Dániel Bereczki.

**Software:** Balázs Dobi.

**Supervision:** András Ajtay, Ferenc Oberfrank, Dániel Bereczki.

**Validation:** András Ajtay, Ferenc Oberfrank, Balázs Dobi, Dániel Bereczki.

**Writing – original draft:** Szabolcs Szatmári, Jr.

**Writing – review & editing:** Szabolcs Szatmári, Jr, Balázs Dobi, Dániel Bereczki.

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
