## [Decision Letter · Decision Letter 0]

5 May 2020

PONE-D-20-03850

The prevalence of psychiatric symptoms before the diagnosis of Parkinson’s disease in a nationwide cohort: a comparison to patients with cerebral infarction

PLOS ONE

Dear Dr. Szatmari Jr,

Thank you for submitting your manuscript to PLOS ONE. After careful consideration, we feel that it has merit but does not fully meet PLOS ONE’s publication criteria as it currently stands. Therefore, we invite you to submit a revised version of the manuscript that addresses the points raised during the review process.

Two reviewers addressed several major and minor concerns about your manuscript. Please revise your manuscript carefully.

We would appreciate receiving your revised manuscript by Jun 19 2020 11:59PM. To enhance the reproducibility of your results, we recommend that if applicable you deposit your laboratory protocols in protocols.io, where a protocol can be assigned its own identifier (DOI) such that it can be cited independently in the future. For instructions see: http://journals.plos.org/plosone/s/submission-guidelines#loc-laboratory-protocols

We look forward to receiving your revised manuscript.

Kind regards,

Kenji Hashimoto, PhD

Academic Editor

PLOS ONE

Journal Requirements:

Reviewers' comments:

Reviewer's Responses to Questions

**Comments to the Author**

1. Is the manuscript technically sound, and do the data support the conclusions?

Reviewer #1: Yes

Reviewer #2: Partly

2. Has the statistical analysis been performed appropriately and rigorously? 

Reviewer #1: No

Reviewer #2: Yes

3. Have the authors made all data underlying the findings in their manuscript fully available?

Reviewer #1: Yes

Reviewer #2: Yes

4. Is the manuscript presented in an intelligible fashion and written in standard English?

Reviewer #1: Yes

Reviewer #2: Yes

5. Review Comments to the Author

Reviewer #1: A short but succinct, relevant and well written result on the rates of occurrence of neuropsychiatric symptoms prior to treatment for PD relative to an appropriate control group. An interesting addition to the PD literature and good use of an administrative data collection.

The statistical analysis presented is correct, however it is not sufficiently complete. All results are quoted without estimates of relative risk and their confidence intervals, with p values reported on their own. While the classical relative risks will be a bit "back to front" in this data, as it provides estimates of the risk of NPS prior to PD relative to the risk of NPS prior to ICL, nevertheless they do quantify the results meaningfully. So estimates of effect size should be included, perhaps added to table 2.

Similarly, using regression to control for age and gender strengthens the results, however these "adjusted " p values should be provided for each comparison, including for the NPS diagnoses prior to PD/ICL daignosis. It would further aid completeness if the authors could comment on the regressions, what was the fit like and any notable effects of age or gender? The authors should consider using the same regression technique for all results, with and without adjustment for age and gender and with and without specific prior NPS diagnoses.

There is a minor multiple testing issue however it is clear that the sample size is more than sufficient to establish the results beyond the usual multiple testing requirements (for example the Bonferroni level) and it would suffice if the authors mentioned this as part of their methodology.

Reviewer #2: Several suggestions are listed behind.

1. Authors should introduce NHIF to readers in the method section. Does NHIF only include the data from all specialist outpatient and inpatient services of all hospitals covering the whole population of Hungary? Does it mean NHIF not include the data from non-specialist outpatient and inpatient services? How about GP? Is it a specialist? Whether PD may be underestimated based on this database needs further explanation in the method or study limitation.

2. What kind of specialist? Neurologists only? Or other specialists are included?

3. For NPS, “For more conservative case ascertainment the F00-F99 diagnoses had to be established at least once in an outpatient or inpatient psychiatric care service.” I think “at least once” is very not reliable. Following the definition of PD above, “at least twice” and “given by related specialists” are needed. For example, anxiety disorder is defined by specific ICD-10 code at least twice given by psychiatrists. Otherwise, the validity is too low.

4. In addition, is NPS defined as “F00-F99 diagnoses”? Is it logical clinically because many ICD diagnoses in F00-F99 are not regarded as NPS anymore, such as Mental retardation, substance, personality….? Authors should specifically point out which ICD codes they want to study with the related references. But, based on authors’ title “The prevalence of psychiatric symptoms before the diagnosis of Parkinson’s disease in a nationwide cohort: a comparison to patients with cerebral infarction”, so I think it may be not very logically relevant to link majority of those psychiatric diagnoses with NPS in their introduction and discussion. So, authors should choose their study and writing strategy for what exact they want to study, and make the manuscript relevant between text and results.

5. “For the control group we chose all patients with at least one ischemic cerebrovascular lesion (ICL) diagnosis (ICD-10, codes I63 or I64) between 2004 and 2016.” Again, authors should explain their logic to find ICL group as control group first. And, “at least once” is very low-valid.

6. “…were excluded as well as patients who refilled any antiparkinsonian drug (APD) before their first PD or ICL diagnosis. Antiparkinsonian agents are coded N04 in accordance with the Anatomical Therapeutic Chemical Classification (ATC) system, patients refilling N04A-anticholinergic agents as well as N04B-dopaminergic agents were excluded.” These two sentences are confusing for the readers. Authors need revision. Does it mean N04A-anticholinergic agents as well as N04B-dopaminergic agents are not included in Antiparkinsonian agents that are excluded from the two groups? They are permitted for use before enrollment date.

7. “Our analysis could evaluate prescriptions of N04 ATC drugs refilled at pharmacies only from 2010 onwards and had no access to data on inpatient medication use in hospitals, therefore pharmacological data in this regard are limited.” Based on this limitation, sensitivity analysis with exclusion of data < 2010 should be done.

8. “their F00-F99 diagnosis established in psychiatric care at least once.” Again, the validity is very low and non-specific.

Thanks.

6. PLOS authors have the option to publish the peer review history of their article (what does this mean?). If published, this will include your full peer review and any attached files.

Reviewer #1: Yes: John F Pearson

Reviewer #2: No

---

## [Author Response · Author response to Decision Letter 0]

24 Jun 2020

REPLY TO REVIEWERS’ COMMENTS

Reviewer 1: A short but succinct, relevant and well written result on the rates of occurrence of neuropsychiatric symptoms prior to treatment for PD relative to an appropriate control group. An interesting addition to the PD literature and good use of an administrative data collection.

We thank Reviewer 1 for evaluating and supporting our manuscript.

Reviewer 1 Q1: The statistical analysis presented is correct, however it is not sufficiently complete. All results are quoted without estimates of relative risk and their confidence intervals, with p values reported on their own. While the classical relative risks will be a bit "back to front" in this data, as it provides estimates of the risk of NPS prior to PD relative to the risk of NPS prior to ICL, nevertheless they do quantify the results meaningfully. So estimates of effect size should be included, perhaps added to table 2.

OUR RESPONSE: We have now complemented our analysis with logistic regression models, and we have added the resulting odds-ratios and confidence intervals to Table 2.

Reviewer 1 Q2: Similarly, using regression to control for age and gender strengthens the results, however these "adjusted" p values should be provided for each comparison, including for the NPS diagnoses prior to PD/ICL diagnosis. It would further aid completeness if the authors could comment on the regressions, what was the fit like and any notable effects of age or gender? The authors should consider using the same regression technique for all results, with and without adjustment for age and gender and with and without specific prior NPS diagnoses. There is a minor multiple testing issue however it is clear that the sample size is more than sufficient to establish the results beyond the usual multiple testing requirements (for example the Bonferroni level) and it would suffice if the authors mentioned this as part of their methodology.

OUR RESPONSE: We have now conducted regression analysis for all F00-F99 subgroups with and without adjustment and included the results of the mixed effects multiple logistic regression models in the manuscript. We report the effect of age and gender now too (generally we found that females had greater odds compared to males and older patients had lowers odds of preceding psychiatric diagnoses). Logistic regression could not be conducted for the sub-diagnoses of F80-89 and F99 due to lack of relevant diagnoses. The adjusted model for the diagnoses F90-98 also had a relatively low amount of diagnoses to work with, and while the regressions could be conducted, they did not produce any significant effect for the PD/ICL groups. 

Reviewer 2: We thank Reviewer 2 for evaluating and supporting our manuscript.

Reviewer 2 Q1: Authors should introduce NHIF to readers in the method section. Does NHIF only include the data from all specialist outpatient and inpatient services of all hospitals covering the whole population of Hungary? Does it mean NHIF not include the data from non-specialist outpatient and inpatient services? How about GP? Is it a specialist? Whether PD may be underestimated based on this database needs further explanation in the method or study limitation.

OUR RESPONSE: We completed the NHIF description with more details in the methods section. We completed the discussion with the limitation of PD underestimation based on this database. 

Reviewer 2 Q2: What kind of specialist? Neurologists only? Or other specialists are included?

OUR RESPONSE: The NHIF database includes data of all specialist also, not only neurologists. GP data are not included. 

Reviewer 2 Q3: For NPS, “For more conservative case ascertainment the F00-F99 diagnoses had to be established at least once in an outpatient or inpatient psychiatric care service.” I think “at least once” is very not reliable. Following the definition of PD above, “at least twice” and “given by related specialists” are needed. For example, anxiety disorder is defined by specific ICD-10 code at least twice given by psychiatrists. Otherwise, the validity is too low.

OUR RESPONSE: We did a full re-analysis including patients with F00-F99 diagnoses established in psychiatric care at least twice to strengthen the validity. The results were modified accordingly however the final conclusion did not change. 

Reviewer 2 Q4: In addition, is NPS defined as “F00-F99 diagnoses”? Is it logical clinically because many ICD diagnoses in F00-F99 are not regarded as NPS anymore, such as Mental retardation, substance, personality….? Authors should specifically point out which ICD codes they want to study with the related references. But, based on authors’ title “The prevalence of psychiatric symptoms before the diagnosis of Parkinson’s disease in a nationwide cohort: a comparison to patients with cerebral infarction”, so I think it may be not very logically relevant to link majority of those psychiatric diagnoses with NPS in their introduction and discussion. So, authors should choose their study and writing strategy for what exact they want to study, and make the manuscript relevant between text and results.

OUR RESPONSE: F00-F99 diagnoses are defined as mental and behavioral disorders. When analyzing the literature we already had a strong suspicion, which psychiatric symptoms were to be the most common but we were curious if there are any other common disorders besides those already known (e.g depression, anxiety). Therefore we analyzed the whole F00-F99 category and discussed ultimately only those which were to be the most common based on our results. 

We changed the term neuropsychiatric symptoms (NPS) in the whole article to psychiatric symptoms (PS) as in the title. We changed NPS to mental and behavioral disorders when defining F00-F99. 

Reviewer 2 Q5: “For the control group we chose all patients with at least one ischemic cerebrovascular lesion (ICL) diagnosis (ICD-10, codes I63 or I64) between 2004 and 2016.” Again, authors should explain their logic to find ICL group as control group first. And, “at least once” is very low-valid.

OUR RESPONSE: Patients with ICL diagnosis were chosen for control group because the age group is somewhat similar to PD and includes large number of patients. 

We did a full re-analysis including patients with ICL with at least two ICL diagnoses to strengthen the validity. The results were modified accordingly however the final conclusion did not change. 

Reviewer 2 Q6: “…were excluded as well as patients who refilled any antiparkinsonian drug (APD) before their first PD or ICL diagnosis. Antiparkinsonian agents are coded N04 in accordance with the Anatomical Therapeutic Chemical Classification (ATC) system, patients refilling N04A-anticholinergic agents as well as N04B-dopaminergic agents were excluded.” These two sentences are confusing for the readers. Authors need revision. Does it mean N04A-anticholinergic agents as well as N04B-dopaminergic agents are not included in Antiparkinsonian agents that are excluded from the two groups? They are permitted for use before enrollment date.

OUR RESPONSE: We agree with Reviewer 2, the above mentioned sentence is confusing for the readers. We reformulated and simplified this statement for better understanding. 

Reviewer 2 Q7: “Our analysis could evaluate prescriptions of N04 ATC drugs refilled at pharmacies only from 2010 onwards and had no access to data on inpatient medication use in hospitals, therefore pharmacological data in this regard are limited.” Based on this limitation, sensitivity analysis with exclusion of data < 2010 should be done.

OUR RESPONSE: We have now conducted a sensitivity analysis where we only kept the data of such psychiatric diagnoses where medication information was also available. We found that the effects and significance are much the same in nearly all models as in the original models, but due to the decreased sample size the effect of PD/ICL diagnosis did not reach significance in four models (F10-19, F50-59, F60-69, F70-79). We have mentioned this in the manuscript. 

Reviewer 2 Q8: “their F00-F99 diagnosis established in psychiatric care at least once.” Again, the validity is very low and non-specific.

OUR RESPONSE: We did a full re-analysis including patients with F00-F99 diagnoses established in psychiatric care at least twice to strengthen the validity. The results were modified accordingly however the final conclusion did not change.

---

## [Decision Letter · Decision Letter 1]

14 Jul 2020

The prevalence of psychiatric symptoms before the diagnosis of Parkinson’s disease in a nationwide cohort: a comparison to patients with cerebral infarction

PONE-D-20-03850R1

Dear Dr. Szatmari Jr,

We’re pleased to inform you that your manuscript has been judged scientifically suitable for publication and will be formally accepted for publication once it meets all outstanding technical requirements.

Kind regards,

Kenji Hashimoto, PhD

Section Editor

PLOS ONE

Additional Editor Comments (optional):

Reviewers' comments:

Reviewer's Responses to Questions

**Comments to the Author**

1. If the authors have adequately addressed your comments raised in a previous round of review and you feel that this manuscript is now acceptable for publication, you may indicate that here to bypass the “Comments to the Author” section, enter your conflict of interest statement in the “Confidential to Editor” section, and submit your "Accept" recommendation.

Reviewer #2: All comments have been addressed

2. Is the manuscript technically sound, and do the data support the conclusions?

Reviewer #2: Yes

3. Has the statistical analysis been performed appropriately and rigorously? 

Reviewer #2: Yes

4. Have the authors made all data underlying the findings in their manuscript fully available?

Reviewer #2: Yes

5. Is the manuscript presented in an intelligible fashion and written in standard English?

Reviewer #2: Yes

6. Review Comments to the Author

Reviewer #2: All questions are well answered and addressed. Revised manuscript is good. I have no further comment. Thanks.

7. PLOS authors have the option to publish the peer review history of their article (what does this mean?). If published, this will include your full peer review and any attached files.

Reviewer #2: No

---

## [Editor Report · Acceptance letter]

23 Jul 2020

PONE-D-20-03850R1 

The prevalence of psychiatric symptoms before the diagnosis of Parkinson’s disease in a nationwide cohort: a comparison to patients with cerebral infarction 

Dear Dr. Szatmári Jr:

I'm pleased to inform you that your manuscript has been deemed suitable for publication in PLOS ONE. Congratulations! Your manuscript is now with our production department. 

Kind regards, 

on behalf of

Prof. Kenji Hashimoto 

Section Editor

PLOS ONE